# ECG Instruction Tuning on Multimodal LLMs for Report Generation: Benchmark and Evaluation

## Abstract

Electrocardiogram (ECG) is the primary non-invasive diagnostic tool for monitoring cardiac conditions and is crucial in assisting clinicians. Recent studies have concentrated on classifying cardiac conditions using ECG data but have overlooked ECG report generation, which is time-consuming and requires clinical expertise. To automate ECG report generation and ensure its versatility, we propose the **M**ultimodal **E**CG **I**nstruction **T**uning (**MEIT**) framework, the *first* attempt to tackle ECG report generation with LLMs and multimodal instructions. To facilitate future research, we establish a benchmark to evaluate MEIT with various LLMs backbones across two large-scale ECG datasets. Our approach uniquely aligns the representations of the ECG signal and the report, and we conduct extensive experiments to benchmark MEIT with nine open-source LLMs using more than 800,000 ECG reports. MEIT's results underscore the superior performance of instruction-tuned LLMs, showcasing their proficiency in *quality report generation*, *zero-shot capabilities*, *resilience to signal perturbation*, and *alignment with human expert evaluation*. These findings emphasize the efficacy of our **MEIT**[1] framework and its potential for real-world clinical application.

## 1 Introduction

Electrocardiogram (ECG) is the primary mechanism for heart disease diagnosis. Cardiologists read and interpret these ECG recordings to manually generate comprehensive ECG reports for heart disease diagnosis, which is a complex and time-consuming process. Recently, AI models have been developed to facilitate ECG data analysis for the task of classification (Hu et al., 2023; Liu et al., 2023a; 2024). Despite these efforts, the automatic generation of reports from ECG recordings still needs to be explored. Unlike other AI-empowered medical report generation applications (e.g., radiology reports), the primary challenge for ECG report generation stems from the distinct nature of ECG content. ECG reports, often comprising brief phrases that summarize signal patterns, contrast with detailed anatomical descriptions in radiology reports. The difference in the content and semantic interpretation between imaging and ECG data complicates the direct application of radiology-focused methods to ECG reports. Furthermore, there is still a lack of comprehensive benchmarks for evaluating the performance of ECG report generation.

To tackle these challenges, we introduce MEIT, a **M**ultimodal **E**CG **I**nstruction **T**uning framework that extends the capabilities of LLMs in the cardiology context to generate ECG reports using ECG recordings and human instructions. Inspired by the versatility of LLMs (Achiam et al., 2023; Touvron et al., 2023a; Wan et al., 2023; Wang et al., 2024a;b) in handling diverse language tasks simultaneously, we develop a specialized instruction tuning process for ECG report generation. MEIT aligns human instructions with ECG recordings, enabling LLMs to generate clinically relevant reports and exhibit zero-shot report generation capabilities under domain transfer scenarios across various continents and data collection devices. Specifically, leveraging publicly available ECG datasets, we construct a multimodal instruction dataset including ECG records, human instructions, and paired reports. Then, we propose an effective and efficient attention-based fusion method to integrate ECG and text representations in the latent space. This enables LLMs to understand ECG signals for report

---

[1]All data and code will be released upon acceptance.

generation without introducing additional training parameters in the attention layer. In addition to the ECG report generation approach, we introduce a comprehensive benchmark for ECG report generation evaluation, utilizing two datasets with 20K and 800K ECG-report pairs, respectively, across four evaluation tasks: report generation quality, zero-shot learning across datasets, robustness analysis in the face of ECG signal perturbation, and alignment with human expert evaluation. Utilizing the ECG report evaluation benchmark, we assess the proposed approach across ten open-source LLMs. The results demonstrate **(1)** the superior performance of MEIT in ECG report generation and the effective learning and alignment of ECG representations; **(2)** the effective transferability of LLMs under the MEIT framework in domain transfer scenarios.

To summarize, our primary contribution is the MEIT framework, a novel approach to automating ECG report generation and evaluation based on LLMs. This framework incorporates a lightweight, attention-based fusion module across various LLM models. Furthermore, we design a new benchmark for ECG report generation, which contains four evaluation tasks. Our evaluations showcase the enhanced capabilities of instruction-tuned LLMs in generating ECG reports, highlighting the transferability in zero-shot tests, robustness against data perturbations, and alignment with human expert evaluation. MEIT paves the way for future advances in automated ECG report generation and methodological innovations in integrating biomedical signals into LLMs.

## 2 RELATED WORK

**Medical Report Generation.** Our work is highly related to the domain of medical report generation. Existing works on medical report generation dominantly focus on medical images, where three categories of techniques have been proposed: (1) Template Selection and Generation, highlighted by HRGR (Li et al., 2018) and CMAS (Jing et al., 2017); (2) Data Integration and Coherence, as seen in PPKED (Liu et al., 2021) and CA (Ma et al., 2021); (3) Cross-Modal Alignment, with efforts like (Chen et al., 2022; Qin and Song, 2022). However, these methods are designed for medical images and face challenges when applied to ECG data due to its unique temporal and waveform characteristics. In contrast, we propose a new approach and benchmark specifically tailored for ECG report generation, effectively addressing these challenges.

**Instruction Tuning.** Our work is also related to instruction tuning. Instruction tuning (Zhang et al., 2023; Wang et al., 2023) boosts zero-shot learning in LLMs for new tasks using instructions. Notable models like InstructGPT (Ouyang et al., 2022), FLAN-PaLM (Chung et al., 2022), and Alpaca (Taori et al., 2023) fine-tune with instruction data through various methods, including human feedback. Similarly, multimodal models such as LLaVA (Liu et al., 2023b), MiniGPT-4 (Zhu et al., 2023), and AnyMAL (Moon et al., 2023) benefit from multimodal instructions for enhanced learning. However, these methods are designed for natural images and cannot be directly applied to ECG signals, which have different characteristics and complexities. Furthermore, instruction tuning for medical signals, especially ECG, remains largely unexplored. In contrast, we propose a novel instruction-tuning framework and benchmark specifically for ECG report generation, addressing this critical gap.

**LLMs for ECG.** Only a few research efforts have focused on utilizing LLMs for ECG signals (Liu et al., 2023c; Qiu et al., 2023; Yu et al., 2023a). In particular, studies such as (Liu et al., 2023c; Yu et al., 2023a) convert ECG signals into text features before feeding them into LLMs, bypassing the original signal data. However, this method overlooks important modality-specific patterns in the signals. Furthermore, these studies focus solely on disease classification from ECG data and do not address medical report generation. Recently, Yu et al. (2023b; 2024) proposes zero-shot ECG diagnosis using LLMs combined with retrieval-augmented generation, significantly improving diagnostic with limited medical data. In contrast, (Qiu et al., 2023) attempts to generate ECG reports by using handcrafted ECG features as input. However, their code and models are not publicly available and focus on classification issues, making direct comparisons difficult. In this work, we propose a new instruction-tuning benchmark and framework that directly utilizes ECG signals for medical signal understanding and report generation, addressing the limitations of prior approaches.

## 3 MEIT

### 3.1 PRELIMINARIES

Electrocardiogram (ECG) measures the electrical activity of an individual's heart over time. An ECG recording typically contains a 12-lead multivariate time series, which acts as a 12-dimensional

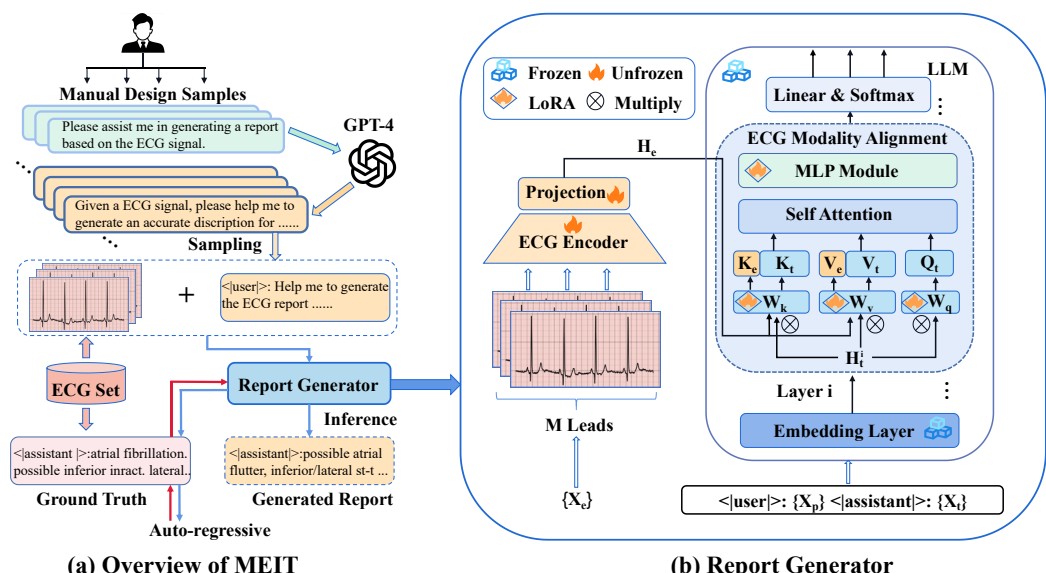

**(a) Overview of MEIT**          **(b) Report Generator**

Figure 1: (a) Overview of MEIT; (b) Illustration of model architecture for ECG **Report Generation**. $\mathbf{K}_e$ and $\mathbf{V}_e$ refer to linear projection of $\mathbf{H}_e$ by multiplying shared $\mathbf{W}_k$ and $\mathbf{W}_v$ in the attention layer.

sequence of embeddings. The ECG signal offers a comprehensive view, encompassing both spatial and temporal aspects of cardiac function. ECG leads can be categorized into six limb leads (i.e., I, II, III, aVR, aVL, and aVF) to monitor arms and legs, providing frontal plane views, and six precordial leads (i.e., V1, V2, V3, V4, V5, and V6) to monitor chest, showing horizontal plane views. We denote an ECG recording as $\mathbf{X}_e \in \mathcal{R}^{M \times T}$, where $M$ represents the number of leads, and $T$ is signal length. Each ECG recording is associated with an ECG report $\mathbf{X}_t$ for description. Thus, we denote each ECG pair as $\{\mathbf{X}_e, \mathbf{X}_t\}$. More details on visualization can be found in the appendix A.7.

## 3.2 FRAMEWORK OVERVIEW

Figure 1 (a) illustrates the proposed MEIT framework. First, we extract and preprocess the ECG signals and corresponding ground truth reports from the ECG dataset to construct the ECG instruction data, which includes instruction prompts, ECG signals, and ground truth. The core steps of this process are detailed in Section 3.3. Next, during the ECG instruction tuning, the processed ECG instruction data is fed into the Report Generator, as shown in Figure 1 (b), for training using an auto-regressive approach. During inference, the instruction prompts and ECG signals are input into the Report Generator to generate professional ECG reports. Next, we describe each component.

## 3.3 DATA CURATION

Given an ECG signal $\mathbf{X}_e$, our goal during inference is to generate an ECG report using an instruction prompt. For instance, the prompt can be "*Given the ECG signal embeddings, please help me generate an accurate description for this ECG signal embeddings:* ". To achieve this goal, during the training phase, we aim to create instruction tuning data to generate a response $\hat{\mathbf{X}}_t$ that aligns semantically with the ground truth $\mathbf{X}_t$. In addition, since we cannot predict the exact instruction prompt that users will use, we need to ensure that our report generation process is robust enough to handle different prompts. To address this challenge, we manually design some prompt samples, then utilize GPT-4 (Achiam et al., 2023) to generate a set of prompts by rephrasing, as shown in Figure 1. Then, we randomly select one instruction prompt $\mathbf{X}_p$ from the prompt set and create a general instruction-following template: `<|user|>`: $\{\mathbf{X}_p, \mathbf{X}_e\}$`<|assistant|>`: $\{\mathbf{X}_t\}$ ``, where `<|user|>` and `<|assistant|>` are added special tokens for tokenizer, `` is a stop sign for each response. This approach ensures that the generated response conveys the same meaning as the ground truth and remains adaptable to different instruction prompts. Following this strategy, we construct the ECG instruction data using a MIMIC-IV-ECG (Gow et al.) dataset that contains 800K annotated data and a 20K dataset PTB-XL (Wagner et al., 2020). The ECG instruction data samples are shown in Appendix A.7.

## 3.4 REPORT GENERATION

In MEIT, the multimodal ECG report generation model decodes ECG signals end-to-end to generate ECG reports. The architecture is illustrated in Figure 1. Specifically, the report generation model directly encodes an entire ECG-signal $\mathbf{X}_e \in \mathcal{R}^{M \times T}$ into latent embeddings and integrate it with the language embeddings with modality alignment, and then autoregressively generate the ECG report. Next, we detail each component of the report generation model.

**ECG Encoder.** Since the ECG signal is of high resolution in the temporal domain, it is vital to efficiently extract temporal features per lead before interaction with semantic embeddings inside the LLM backbone. Our default ECG encoder $\mathcal{F}_e(\cdot)$ consists of temporal convolution blocks to encode each ECG signal into embeddings. Specifically, each temporal convolution block comprises several 1-D convolution layers, batch normalization layers, and ReLU activation layers, followed by average pooling. This design allows us to effectively capture temporal dependencies and reduce the complexity of the signal representations, ensuring that the model can quickly learn important temporal features efficiently. To further align the output dimension with the head dimension of the LLM backbone $\mathcal{F}_l(\cdot)$, we employ a non-linear projection layer $\mathcal{P}_e(\cdot)$ to generate the ECG embeddings:

$$\mathbf{H}_e = \mathcal{P}_e \left( \mathcal{F}_e \left( \mathbf{X}_e \right) \right), \tag{1}$$

where $\mathbf{H}_e \in \mathcal{R}^{D_h}$, $D_h$ has the same dimension as the multi-head attention layers of LLMs. Note that our default ECG encoder is lightweight and is able to learn temporal patterns of signals without a long training period. More details about ECG Encoder are illustrated in Appendix A.2.

**ECG Modality Alignment.** We introduce an ECG modality alignment strategy to guide the LLMs in aligning ECG signal data with corresponding textual outputs. This approach is detailed in Figure 1 (b). Specifically, given the ECG embeddings $\mathbf{H}_e$, the alignment strategy incorporates $\mathbf{H}_e$ with the current hidden state $\mathbf{H}_t^i$ generated from previous $i - 1$-th layer of the LLM backbone $\mathcal{F}_l(\cdot)$ for next-token prediction task. Here $\mathbf{H}_t^i$ is defined as:

$$\mathbf{H}_t^i = \mathcal{F}_l^{i-1} \left( [\mathbf{X}_p, \mathbf{X}_t] \right), \tag{2}$$

where $i$ is the current layer index. Traditional gated-attention fusion in Flamingo (Alayrac et al., 2022), Memorizing Transformer (Wu et al., 2022), and G-MAP (Wan et al., 2022), or Q-former in BLIP-2 (Li et al., 2023) that requires additional trainable parameters and designed for complex multi-stage alignment of rich semantic information (e.g., images). Different from them, our method provides a lightweight concatenated-fusion alignment strategy tailored to the embeddings of ECG signals, enabling efficient learning of ECG semantic features via directly injecting the ECG embeddings with language context in the self-attention, while preventing potential catastrophic forgetting of general knowledge in LLMs. In our approach, each attention layer combines $\mathbf{H}_e$, generated from the ECG encoder and projector as a prefix condition, with $\mathbf{H}_t^i$, derived from the preceding layer. The fusion process is as follows:

$$\text{Self-Attn} \left( \mathbf{H}_e, \mathbf{H}_t^i \right) = [\text{head}_1, \ldots, \text{head}_k] \mathbf{W}_o, \tag{3}$$

where $k$ represents the number of attention heads, and $\mathbf{W}_o$, a matrix in $\mathcal{R}^{kD_h \times D_m}$, serves as the projection matrix with $D_m$ denoting the hidden size of the LLM backbone. We replicate $\mathbf{H}_e$ for each head $k$ times, merging the ECG and language features in the sequence dimension. This is achieved through a shared projection of keys and values for each pattern. The fusion is then articulated as:

$$\mathbf{K}_{m,j} = [\mathbf{K}_{e,j}, \mathbf{K}_{t,j}]^\top, \mathbf{V}_{m,j} = [\mathbf{V}_{e,j}, \mathbf{V}_{t,j}], \tag{4}$$

$$\text{head}_j = \text{Softmax} \left( \frac{\mathbf{Q}_{t,j} \mathbf{K}_{m,j}}{\sqrt{D_h}} \right) \mathbf{V}_{m,j}, \tag{5}$$

where $\mathbf{Q}_{t,j} = \mathbf{H}_{t,j}^i \mathbf{W}_{q,j}$, $\mathbf{K}_{e,j} = \mathbf{H}_e \mathbf{W}_{k,j}$, and $\mathbf{K}_{t,j} = \mathbf{H}_{t,j}^i \mathbf{W}_{k,j}$, with a similar notation for $\mathbf{V}_{e,j} = \mathbf{H}_e \mathbf{W}_{v,j}$ and $\mathbf{V}_{t,j} = \mathbf{H}_t^i \mathbf{W}_{v,j}$. Concatenation is denoted by $[\cdot]$, and $\mathbf{K}_{m,j}$ and $\mathbf{V}_{m,j}$ symbolize the amalgamated features of query and key. $\mathbf{W}_{q,j}$, $\mathbf{W}_{k,j}$, and $\mathbf{W}_{v,j}$ in $\mathcal{R}^{D_h \times D_h}$ represent the projection matrices for query, key, and value for each head $j$, respectively. Our model's design allows for the efficient fusion of two modalities through causal attention, facilitating conditional generation without the need for additional parameter updates to align the ECG modality. Ablation studies comparing with other fusion methods demonstrate the effectiveness and efficiency of our proposed lightweight alignment strategy. More comparisons about ECG modality alignment and other fusion approaches are illustrated in Table 6.

## 3.5 Instruction Tuning

As described in Section 3.3, we have converted ECG-text pairs into a chat-bot style instruction format: `<|user|>`: $\{\mathbf{X}_p, \mathbf{X}_e\}$`<|assistant|>`: $\{\mathbf{X}_t\}$``. During instruction tuning, we compute autoregressive loss only on tokens after response tokens `<assistant>`, and use label loss masking to finetune the model, where we mask all tokens belonging to $\mathbf{X}_p$ and $\mathbf{X}_e$. To save computational resources and accelerate the convergence of instruction tuning, we use LoRA (Hu et al., 2021) adapters for all linear layers of the LLM backbone $\mathcal{F}_l$ and freeze its backbone. Subsequently, given a sequence of ECG instruction data, we compute the probability of the target response $\mathbf{X_t}$ as an autoregressive function:

$$p\left(\mathbf{X}_t \mid \mathbf{X}_p, \mathbf{X}_e\right) = \prod_{i=j}^{L} p_{\boldsymbol{\theta}}\left(\mathbf{x}_{t,i} \mid \mathbf{X}_p, \mathbf{X}_e, \mathbf{X}_{t,<i}\right), \tag{6}$$

where $j$ is the start index after `<assistant>`, $\theta$ is the trainable parameters of LoRA and ECG encoder $\mathcal{F}_e$, $\mathbf{X}_{t,<i}$ is the response tokens before the current generation $\mathbf{x}_{t,i}$.

# 4 ECG Report Generation Benchmark

## 4.1 Datasets

**PTB-XL.** The PTB-XL dataset (Wagner et al., 2020) contains $21,837$ clinical 12-lead ECG recordings, each sampled at 500Hz and lasting 10 seconds, collected from $18,885$ patients. Each ECG recording has a corresponding report. We divided this dataset into training, validation, and testing subsets in a 70%:10%:20% ratio, respectively. The human experts double-check all samples in the test set to ensure data quality. As mentioned in Sec 3.3, we reformulate the dataset into the instruction data format.

**MIMIC-IV-ECG.** The MIMIC-IV-ECG dataset (Gow et al.) is currently the largest publicly accessible ECG dataset, containing 800,035 paired samples from 161,352 unique subjects. Similar to PTB-XL, each sample in this dataset includes a raw ECG signal and its corresponding report, with all recordings sampled at 500Hz for 10 seconds. The division of this dataset into training, validation, and testing subsets is 80%:10%:10% ratio. Likewise, we reconstruct this dataset into an ECG instruction data template.

## 4.2 Models

We use nine LLMs based on the peft[2] library, which directly supports LoRA (Hu et al., 2021) to construct the multimodal ECG report generation model described in Section 3.4. These models include GPT-Neo (Black et al., 2021), GPT-NeoX (Black et al., 2022), GPT-J (Wang and Komatsuzaki, 2021), BLOOM (Workshop et al., 2022), OPT (Zhang et al., 2022), LLaMA-1 (Touvron et al., 2023b), LLaMA-2-Instruct (Touvron et al., 2023a), LLaMA-3-Instruct (Touvron et al., 2023a), Mistral (Jiang et al., 2023), and Mistral-Instruct[3], along with two relatively small pre-trained language models (GPT2-Medium and GPT-Large (Radford et al., 2019)) as fundamental baselines.

## 4.3 Evaluation Metrics

We evaluate models using nine metrics: BLEU 1-4 (Papineni et al., 2002), METEOR (Banerjee and Lavie, 2005), ROUGE 1-2 and L (Lin, 2004), CIDEr-D (Vedantam et al., 2015), and BERTScore (Zhang et al., 2019). BLEU and METEOR assess machine translation quality, focusing on accuracy and fluency. ROUGE-L measures sentence fluency and structure, while ROUGE-1 and ROUGE-2 examine uni-gram and bi-gram overlaps. CIDEr-D evaluates the relevance and uniqueness of generated ECG reports against a candidate set, and BERTScore assesses semantic similarity to the ground truth, ensuring content accuracy.

---

[2]https://github.com/huggingface/peft
[3]https://huggingface.co/mistralai/Mistral-7B-Instruct-v0.1

Table 1: Natural language generation metric on MIMIC-IV-ECG. For model size, 'M' denotes the million level, and 'B' denotes the billion level. All checkpoints are downloaded from Hugging Face website. And all models have been fine-tuned using ECG instructions. The  light teal  color indicates the second highest results, and  heavy teal  color indicates the highest results.

| MODELS | SIZE | BLEU-1 | BLEU-2 | BLEU-3 | BLEU-4 | METEOR | ROUGE-L | ROUGE-1 | ROUGE-2 | CIDEr-D |
|---|---|---|---|---|---|---|---|---|---|---|
| GPT2-Medium | 345M | 0.576 | 0.527 | 0.456 | 0.425 | 0.551 | 0.523 | 0.544 | 0.512 | 3.70 |
| GPT2-Large | 774M | 0.614 | 0.563 | 0.490 | 0.476 | 0.595 | 0.571 | 0.585 | 0.538 | 4.21 |
| GPT-Neo | 2.7B | 0.631 | 0.579 | 0.534 | 0.489 | 0.727 | 0.689 | 0.715 | 0.592 | 4.81 |
| GPT-NeoX | 20B | 0.645 | 0.588 | 0.539 | 0.523 | 0.719 | 0.701 | 0.712 | 0.622 | 4.92 |
| GPT-J | 6B | 0.676 | 0.628 | 0.584 | 0.542 | 0.756 | 0.721 | 0.744 | 0.632 | 5.23 |
| BLOOM | 7B | 0.669 | 0.624 | 0.591 | 0.550 | 0.758 | 0.725 | 0.745 | 0.639 | 5.19 |
| OPT | 6.7B | 0.673 | 0.616 | 0.598 | 0.532 | 0.755 | 0.732 | 0.743 | 0.631 | 5.32 |
| LLaMA-1 | 7B | 0.685 | 0.648 | 0.615 | 0.543 | 0.761 | 0.724 | 0.742 | 0.642 | 5.26 |
| Mistral | 7B | 0.697 | 0.659 | 0.611 | 0.571 | 0.763 | 0.740 | 0.765 | 0.658 | 5.48 |
| LLaMA-2-Instruct | 7B | 0.706 | 0.662 | 0.622 | 0.581 | 0.775 | 0.745 | 0.768 | 0.664 | 5.55 |
| Mistral-Instruct | 7B | 0.714 | 0.665 | 0.619 | 0.576 | 0.768 | 0.751 | 0.762 | 0.667 | 5.62 |
| LLaMA-3-Instruct | 8B | 0.733 | 0.686 | 0.648 | 0.610 | 0.799 | 0.773 | 0.795 | 0.686 | 5.78 |

Table 2: Natural language generation metric on PTB-XL. The  light teal  color indicates the second highest results, and  heavy teal  color indicates the highest results.

| MODELS | SIZE | BLEU-1 | BLEU-2 | BLEU-3 | BLEU-4 | METEOR | ROUGE-L | ROUGE-1 | ROUGE-2 | CIDEr-D |
|---|---|---|---|---|---|---|---|---|---|---|
| GPT2-Medium | 345M | 0.329 | 0.278 | 0.254 | 0.232 | 0.441 | 0.391 | 0.561 | 0.433 | 2.12 |
| GPT2-Large | 774M | 0.437 | 0.395 | 0.355 | 0.320 | 0.575 | 0.481 | 0.652 | 0.527 | 3.25 |
| GPT-Neo | 2.7B | 0.474 | 0.449 | 0.398 | 0.373 | 0.602 | 0.486 | 0.674 | 0.595 | 3.70 |
| GPT-NeoX | 20B | 0.469 | 0.453 | 0.417 | 0.399 | 0.620 | 0.553 | 0.688 | 0.622 | 3.58 |
| GPT-J | 6B | 0.485 | 0.452 | 0.428 | 0.405 | 0.656 | 0.550 | 0.662 | 0.613 | 3.72 |
| BLOOM | 7B | 0.491 | 0.462 | 0.427 | 0.415 | 0.665 | 0.580 | 0.678 | 0.605 | 3.80 |
| OPT | 6.7B | 0.502 | 0.477 | 0.431 | 0.418 | 0.662 | 0.568 | 0.669 | 0.624 | 3.94 |
| LLaMA-1 | 7B | 0.514 | 0.485 | 0.465 | 0.430 | 0.678 | 0.588 | 0.682 | 0.613 | 3.97 |
| Mistral | 7B | 0.486 | 0.475 | 0.446 | 0.421 | 0.673 | 0.591 | 0.697 | 0.634 | 3.98 |
| LLaMA-2-Instruct | 7B | 0.515 | 0.484 | 0.469 | 0.439 | 0.675 | 0.594 | 0.698 | 0.624 | 4.05 |
| Mistral-Instruct | 7B | 0.501 | 0.481 | 0.457 | 0.425 | 0.664 | 0.592 | 0.700 | 0.641 | 4.01 |
| LLaMA-3-Instruct | 8B | 0.539 | 0.513 | 0.494 | 0.467 | 0.698 | 0.615 | 0.725 | 0.646 | 4.45 |

## 4.4 TASKS

**Quality of Generated Reports.** This task aims to assess report quality after ECG instruction tuning using 10% of MIMIC-IV-ECG and PTB-XL datasets as the test set. The evaluation examines how closely generated reports match the original's structure and meaning, considering various instructions and ECG inputs. We analyze metrics like BLEU-1 to 4, METEOR, ROUGE 1, 2, L, CIDEr-D, and BERTScore.

**Zero-shot Generalizability.** To explore the generalizability of LLMs in domain transfer scenarios following ECG instruction tuning, we trained the models on 70% of the instruction data from MIMIC-IV-ECG. Following this, we evaluated the models' zero-shot capabilities on the PTB-XL test set. It's important to note that the PTB-XL and MIMIC-IV-ECG datasets originate from different continents—Europe and the United States, respectively—utilizing varied devices and from distinct hospitals, across different time periods. Therefore, we consider these datasets to represent two separate domains. This distinction allows us to use the PTB-XL dataset to gauge our model's performance in zero-shot domain transfer effectively. We used the metrics BLEU-4, METEOR, ROUGE-L, and CIDEr-D because of limited space and calculated their average for model evaluation.

**Signal Perturbation Robustness.** In real-world clinical settings, ECG signals often contain some degree of noise. To evaluate the robustness of MEIT against such noisy interference, we selected 10% of the ECG samples from the MIMIC-IV-ECG test dataset. We then added Gaussian noise to these samples during the models' instruction-based inference process. For this evaluation, we used BLEU-4, METEOR, ROUGE-L, and CIDEr-D as metrics.

**Evaluation of Alignment with Human Expert Annotations.** To evaluate the differences between the reports generated by ECG-instructed LLMs and human expert annotations, we established specific evaluation criteria and utilized closed-source LLMs to conduct a professional assessment of both the generated reports and expert annotations.

# 5 EXPERIMENTS AND ANALYSIS

## 5.1 EXPERIMENTAL SETUP

In this section, we evaluate and benchmark ten open-source decoder-only LLMs using the constructed ECG report generation benchmark. Additionally, we offer a comprehensive analysis of scalability and instruction tuning and present case studies showcasing the generated reports.

**Implementation Details.** Our study utilized PyTorch 2.1, transformers (Wolf et al., 2020), and accelerated on A100 GPUs with LLMs from Hugging Face (Wolf et al., 2019) ranging from 2.7 to 70 billion parameters. For larger models, we used DeepSpeed[4]. The training covered 5 epochs on MIMIC-IV-ECG and PTB-XL with a 2e-5 learning rate and 64 batch size, employing a linear optimizer with a 0.03 warm-up ratio.

Table 3: Semantic similarity between the generated ECG reports and ground truths is measured using BERTScore, denoted as P for Precision, R for Recall, and F-1 for the F-1 Score.

| MODELS | MIMIC-IV-ECG | | | PTB-XL | | |
|---|---|---|---|---|---|---|
| | **P** | **R** | **F-1** | **P** | **R** | **F-1** |
| GPT2-Medium | 0.562 | 0.453 | 0.502 | 0.534 | 0.465 | 0.497 |
| GPT2-Large | 0.657 | 0.574 | 0.613 | 0.625 | 0.553 | 0.586 |
| GPT-Neo | 0.723 | 0.633 | 0.675 | 0.675 | 0.588 | 0.628 |
| GPT-NeoX | 0.719 | 0.638 | 0.676 | 0.654 | 0.579 | 0.614 |
| GPT-J | 0.725 | 0.655 | 0.688 | 0.689 | 0.622 | 0.654 |
| BLOOM | 0.734 | 0.684 | 0.708 | 0.701 | 0.645 | 0.672 |
| OPT | 0.713 | 0.667 | 0.689 | 0.712 | 0.648 | 0.678 |
| LLaMA-1 | 0.752 | 0.697 | 0.723 | 0.725 | 0.657 | 0.689 |
| Mistral | 0.761 | 0.732 | 0.746 | 0.711 | 0.664 | 0.687 |
| LLaMA-2-Instruct | 0.764 | 0.725 | 0.744 | 0.721 | 0.668 | 0.693 |
| Mistral-Instruct | 0.773 | 0.722 | 0.747 | 0.730 | 0.661 | 0.694 |
| LLaMA-3-Instruct | 0.798 | 0.745 | 0.771 | 0.745 | 0.682 | 0.712 |

For text preprocessing, we initially remove all instances of the 'nan' string and sentences that consist solely of numerical values. Subsequently, we discard any samples whose reports contain fewer than 5 tokens. For, ECG encoder, we adopt random initialization. Additionally, the default number of generated prompts from GPT-4 is 256, more training, visualization details about ECG instruction tuning are illustrated in Appendix A.1, Section 5.3, and Appendix A.3.

## 5.2 BENCHMARK TASK EVALUATION

### 5.2.1 QUALITY EVALUATION

**Performance on MIMIC-I V-ECG.** Table 3 and 1 present the results of various types of language encoders $\mathcal{F}_l(\cdot)$ on MIMIC-IV-ECG. The results show that all LLMs perform better than smaller language models (SLMs), such as GPT2-Medium and GPT2-Large, across all evaluation metrics. Notably, from GPT-Neo to Mistral-Instruct, LLM-based backbones achieve

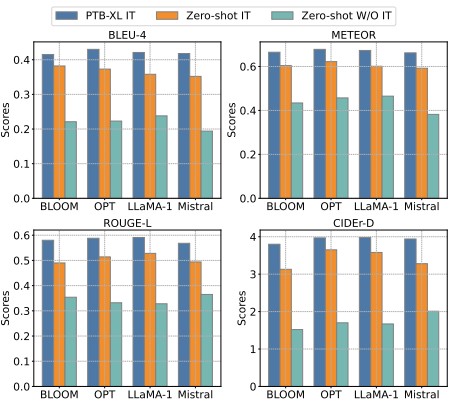

a significant margin over SLMs in all metrics. For instance, compared to GPT2-Large, the METEOR score increases in the range of 0.132 to 0.18 from GPT-Neo to LLaMA-2, and Mistral-Instruct outperforms GPT2-Large with an improvement of 0.18 in the ROUGE-L score and 0.134 in the F-1 of BERTScore. The observed performance underscores the adeptness of LLMs in generalizing from signal data, showcasing enhanced proficiency in aligning ECG signal representations with corresponding textual information. This highlights the significant potential of LLMs in medical signal-to-text generation. Particularly, LLaMA-2-Instruct, Mistral-Instruct, and LLaMA-3-Instruct surpass their counterparts in most evaluative metrics, suggesting that models pre-tuned with general instructions are more adept at learning ECG-text alignment.

Figure 2: Zero-shot performance on PTB-XL dataset. "IT" denotes instruction tuning.

**Performance on PTB-XL.** As shown in Table 2, the models exhibit reduced performance on PTB-XL compared to MIMIC-IV-ECG, which is attributable to the smaller scale of the instruction data in PTB-XL. This underscores the importance of data scale in enhancing instruction-based ECG report generation. Moreover, similar to the MIMIC-IV-ECG results, all LLM-based models show significant improvement over SLMs. Specifically, LLaMA-2 surpasses GPT2-Large by 0.134 in the BLEU-3 metric, while LLaMA-1 achieves a 0.103 improvement in the METEOR score. The

---

[4]https://github.com/microsoft/DeepSpeed

overall experimental results also reveal that Mistral-Instruct, LLaMA-2-Instruct, and LLaMA-3-Instruct are consistently the top two performers across most metrics because of their strong general instruction-following capabilities.

### 5.2.2 ZERO-SHOT EVALUATION IN DOMAIN TRANSFER.

Although both PTB-XL and MIMIC-IV-ECG datasets are time-series data, they differ significantly in several aspects, including population (European vs. American), diverse collection devices, continents (Europe vs. US), protocols, and hospitals. These differences introduce substantial medical domain gaps (Bilheimer and Klein, 2010; Ross et al., 2020). In Figure 2, we present the evaluation of the zero-shot learning capabilities of various LLMs, which is trained on the MIMIC-IV-ECG dataset and then tested on PTB-XL (unseen dataset). The assessed models include BLOOM, OPT, LLaMA-1, and Mistral. Firstly, all selected LLMs undergo instruction tuning on the MIMIC-IV-ECG train set, followed by zero-shot testing on the PTB-XL test set verified by human experts, denoted as ZERO-SHOT IT. We also measure the performance of each model in report generation without prior ECG-specific instruction tuning, denoted as ZERO-SHOT W/O IT. PTB-XL IT represents training on the PTB-XL train set and then evaluated on the PTB-XL test set. Notably, although ZERO-SHOT IT shows a slight degradation compared to PTB-XL IT, the results still indicate a variance in the model's ability to generalize to an unseen dataset with instruction tuning (IT), compared to ZERO-SHOT W/O IT. The involvement of ECG instruction tuning on MIMIC-IV-ECG enables the models to achieve superior zero-shot performance on the unseen PTB-XL dataset, indicating the necessity of instruction tuning in enhancing the models' zero-shot ability on unseen datasets in ECG report generation.

### 5.2.3 ROBUST ANALYSIS WITH PERTURBED ECG SIGNAL.

In a noise stress evaluation (Wang et al., 2019), we added Gaussian noise to ECG signals at signal-to-noise ratios (SNRs) of 0.05, 0.1, 0.15, and 0.2 during testing to assess model robustness. Our experiments utilized four LLM architectures: BLOOM, OPT, LLaMA-1, and Mistral, each trained on clean ECG signals from the MIMIC-IV-ECG training set and tested on corresponding noise-added signals from its test set. The results, illustrated in Figure 3, show a performance decline in all LLMs as SNR decreased, highlighting the significant interference of ECG noise. Furthermore, as shown in Table 1, Mistral also excelled in tests on noise-free datasets, suggesting a synergistic effect between clean and noisy test sets. The results demonstrate Mistral's strong resistance to perturbations. Even

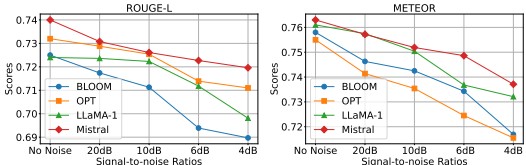

Figure 3: Signal perturbation robustness analysis on various LLMs.

with more severe noise, it maintained robustness regarding ROUGE-L and METEOR metrics. Developing an even more robust framework is a goal for future research.

Table 4: Prompt template used for GPT-4o evaluation. This prompt guided the model's evaluation of generated ECG reports.

| Prompt Template for GPT-4o Evaluation |
|---|
| You are an expert in Electrocardiogram (ECG) text evaluation. Your task is to assess the quality of a generated ECG report by comparing it to a real, expert-annotated ECG report.
**Generated ECG Report**: {Generated_Report}
**Real ECG Report**: {Real_Report}
Please evaluate the generated report based on the following criteria: |
| 1. **Medical Terminology Accuracy**: Does the generated report use correct and appropriate ECG signal terms?
2. **Logical Consistency**: Is the information presented in a logical and medically sound order?
3. **Completeness**: Does the report include all necessary details that would be present in a real ECG report, such as heart rhythm, rate, and any abnormalities?
4. **Diagnostic Accuracy**: Are the diagnoses and interpretations in the generated report accurate and consistent with the expert-annotated report?
Please provide a detailed analysis and score each criterion on a scale of 1 to 5 (1 = Poor, 5 = Expert-Level). |

### 5.2.4 EVALUATION OF ALIGNMENT WITH HUMAN EXPERT ANNOTATIONS.

We conducted an evaluation of model-generated ECG reports from ECG instruction-tuned versions of LLaMA-2 and LLaMA-3 against 500 ground-truth reports, meticulously annotated by human

Table 5: Evaluation results of LLaMA-2-Instruct and LLaMA-3-Instruct against human expert-annotated ground-truth reports. Each dimension is scored on a scale of 1 to 5.

| Model | Medical Terminology Accuracy | Logical Consistency | Completeness | Diagnostic Accuracy |
|---|---|---|---|---|
| LLaMA-2-Instruct | 4.25 | 4.11 | 3.72 | 3.60 |
| LLaMA-3-Instruct | **4.52** | **4.38** | **4.01** | **3.98** |

medical experts. These test annotated data were randomly sampled from the PTB-XL dataset , with all selected reports carefully reviewed and validated by human experts. Each model-generated report was compared with these expert-annotated reports using gpt-4o[5], which assessed quality across four dimensions: **Medical Terminology Accuracy**, **Logical Consistency**, **Completeness**, and **Diagnostic Accuracy**, on a scale of 1 to 5. To evaluate these reports, we employed the following prompt template, which guided GPT-4o's scoring process across the defined dimensions, as shown in Table 4. This prompt template ensures that GPT-4o evaluates the reports in a structured and consistent manner, highlighting both strengths and weaknesses of the model-generated reports in comparison to human expert annotations. The results indicate that the LLaMA-3 model, with an average Diagnostic Accuracy score of 3.85, closely matches the quality of the human expert annotations, whereas the LLaMA-2 model scored 3.60. This evaluation underscores the effectiveness of using human expert annotations from the PTB-XL (Wagner et al., 2020) dataset as a rigorous benchmark for assessing the models' ability to generate clinically reliable ECG reports.

Table 6: Performance comparison of the proposed concatenated-fusion method and other mainstream fusion variants. We evaluate these methods on the MIMIC-IV-ECG dataset, using BLEU-4, METEOR, ROUGE-L, and CIDEr-D metrics. We take LLaMA-1 7B as the LLM backbone here. heavy teal color indicates the highest results.

| FRAMEWORK | METHOD | BLEU-4 | METEOR | ROUGE-L | CIDEr-D |
|---|---|---|---|---|---|
| LLaVA | Straightforward input | 0.529 | 0.737 | 0.712 | 4.99 |
| Flamingo | Trainable cross-attention | 0.527 | 0.768 | 0.715 | 5.11 |
| MEIT | Concatenated-fusion | 0.543 | 0.761 | 0.724 | 5.26 |

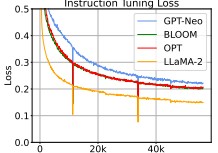 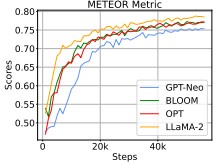 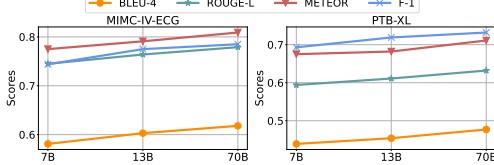

Figure 4: Visualizations of instruction tuning loss and METEOR score.

Figure 5: Model scaling performance on MIMIC-IV-ECG and PTB-XL.

## 5.3 ANALYSIS

**Instruction Tuning Visualization.** Figure 4 compares the convergence curves of the instruction tuning loss and the METEOR score between GPT-Neo (2.7B), BLOOM (7B), OPT (6.7B), and LLaMA-2 (7B) on the MIMIC-IV-ECG train and validation datasets. We observe that larger models with more parameters can converge to a more minor loss and achieve higher performance on the METEOR score. Notably, an increase in model size correlates with higher performance and lower loss, suggesting that larger models have the potential for better performance.

**Analysis of ECG Modality Alignment.** To study the effectiveness of our proposed concatenated-fusion method for ECG modality alignment, we compare it with other fusion approaches such as direct input in LLaVA (Liu et al., 2023b) and additional trainable cross-attention layer in Flamingo (Alayrac et al., 2022). For straightforward input, we follow the design of LLaVA by directly concatenating the ECG encoder's output embeddings with the sentence's embeddings before inputting them into the LLM backbones. For the second comparison method, we follow Flamingo by adding a trainable cross-attention layer within the attention block. From Table 6, we observe that the Concatenated-fusion method outperforms the trainable cross-attention method of Flamingo in most metrics and is consistently superior to the Straightforward input method of LLaVA. Consequently, the concatenated

[5]https://platform.openai.com/docs/models/gpt-4o

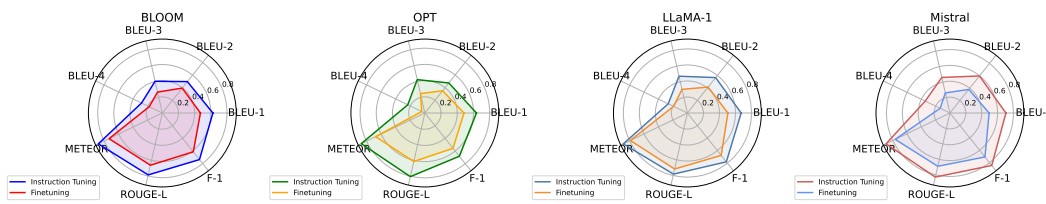

Figure 6: Ablation Study of ECG Instruction Tuning on MIMIC-IV-ECG Dataset.

fusion is more effective for the LLM backbone's alignment with fine-grained ECG patterns without necessitating additional trainable parameters.

**Scalability Analysis.** To investigate whether ECG instruction tuning on larger-scale models yields better results, we validated LLaMA-2 models of 7B, 13B, and 70B parameter sizes on both MIMIC-IV-ECG and PTB-XL datasets. As depicted in Figure 5, an upward trend in all evaluation metrics is observed with a gradual increase in model size.

However, it is noteworthy that the gains in performance associated with increasing model size are not particularly significant. For example, the F-1 score for the 70B model on the PTB-XL dataset exhibits a marginal increase of 0.02 over the 13B model. Similarly, on the MIMIC-IV-ECG dataset, the 70B model's F-1 score is only 0.01 higher than that of the 13B model. Therefore, we conjecture that enhancing both data scale and model size concurrently is necessary to achieve superior performance (Wei et al., 2022).

**Ablation Study on ECG Instruction Tuning.** We conducted an ablation study to evaluate instruction tuning's impact on aligning ECG signals with report representations. Utilizing LLMs such as BLOOM, OPT, LLaMA-1, and Mistral without instruction tuning, we allowed direct learning from ECG signals. The findings, illustrated in Figure 6, indicate a significant performance drop across all metrics without instruction tuning, particularly in Mistral. This underscores instruction tuning's superiority in enhancing LLMs' generalization to new tasks/data over direct fine-tuning (Ouyang et al., 2022).

**Qualitative Results.** In Figure 7, we randomly select two samples generated by MEIT using LLaMA-2 and Mistral-Instruct as the LLM backbones. The consistent key information, highlighted in blue, indicates that both models have

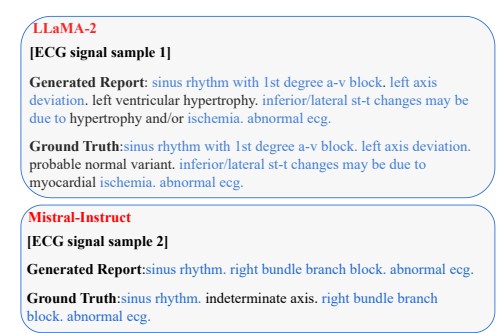

Figure 7: Examples of ECG reports generated by LLaMA-2 and Mistral-Instruct. We highlight the consistent information between the generated reports and the ground truths with blue color.

successfully learned important patterns from the ECG signals. Overall, the models' results align with the ground truth, accurately identifying cardiac abnormalities from the ECG signals. Furthermore, both models provide detailed explanations of abnormal ECG signal details, such as 'ischemia' from sample 1 and 'right bundle branch block' from sample 2. These generated reports demonstrate the efficacy of our method.

## 6 CONCLUSION

In this paper, we introduced MEIT, a new framework for generating instruction-following data to train a multimodal LLM that can produce ECG reports based on human instructions. We also proposed an effective method for aligning ECG and report representations across various open-source LLMs, demonstrating strong performance on both the MIMIC-IV-ECG and PTB-XL datasets across multiple tasks. Additionally, we established a comprehensive benchmark for ECG instruction-following in report generation, providing a standardized evaluation for future research. Although this work primarily focuses on ECG signals, it serves as a foundational step in applying instruction-tuning to biomedical signals. For future research, we aim to extend our framework and benchmark to other medical domains, such as EEG, with the hope of driving further progress in developing more capable medical-signal LLMs.

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

# A  APPENDIX.

## A.1  HYPER-PARAMETERS OF ECG INSTRUCTION TUNING

Table 7: Hyper-parameters of ECG instruction tuning for all LLM backbones.

| Hyperparameters | |
|---|---|
| Mixed precision | bf16 |
| Instruction tuning epochs | 5 |
| LoRA alpha | 64 |
| LoRA rank | 128 |
| LoRA dropout | 0.1 |
| Total batch size | 64 |
| Gradient accumulation | 2 |
| Maximum sequence length | 256 |
| Learning rate | 2e-5, 1e-4 |
| Learning rate Optimizer | AdamW |
| Schedule | linear |
| Warm-up ratio | 0.03 |
| Weight decay | 0.0 |

Table 8: ECG dimension of different language models.

| MODELS | ECG Dimension |
|---|---|
| GPT2-Medium | 64 |
| GPT2-Large | 64 |
| GPT-Neo | 128 |
| GPT-NeoX | 96 |
| GPT-J | 256 |
| BLOOM | 128 |
| OPT | 128 |
| LLaMA-1 | 128 |
| Mistral | 128 |
| LLaMA-2 | 128 |
| Mistral-Instruct | 128 |

In this study, we implement the Low-Rank Adaptation (LoRA) (Hu et al., 2021) technique for efficient fine-tuning, specifically applied to ECG instruction tuning. As detailed in Table 7 provided, we utilize mixed precision at bf16 for enhanced computational efficiency. Our models undergo instruction tuning over 5 epochs, with LoRA parameters set at an alpha of 64 and a rank of 128, accompanied by a dropout rate of 0.1. The total batch size is 64, with a gradient accumulation factor of 2. The maximum sequence length is constrained to 256 tokens. Additionally, we adopt a learning rate with 2e-5 for GPT-NeoX and 1e-4 for the other models, optimized using the AdamW algorithm. The learning rate follows a linear schedule with a warm-up ratio of 0.03. We set the weight decay to 0.0.

Moreover, as shown in Table 8, we detail the ECG embedding dimensions for various language models, highlighting their approach to ECG data encoding. GPT2-Medium and GPT2-Large feature ECG dimensions 64, while GPT-Neo, BLOOM, OPT, LLaMA-1, Mistral, LLaMA-2, and Mistral-Instruct use a dimension of 128. GPT-NeoX employs a dimension of 96, and GPT-J notably uses the largest dimension of 256. These dimensions, reflecting each model's head dimension design, illustrate diverse strategies in ECG data processing across different models.

## A.2  MORE DETAILS OF ECG ENCODER

**Projection Layer** For the design of the projection layer within the ECG encoder, we adopt a non-linear approach similar to CLIP (Radford et al., 2021) and Med-UniC (Wan et al., 2024). Specifically, in our experiments, we employ two consecutive linear layers, each followed by BatchNorm1d[6]. Besides, ReLU serves as the activation function between the two linear layers. The default settings for input and hidden layers dimensions are set to 2048 in our experiment.

**Parameter Size Analysis** To demonstrate the ECG encoder's lightweight design, we analyzed its trainable parameters during instruction tuning and total parameters during inference, using the LLaMA-1 7B model for illustration (Table 10). The analysis reveals the ECG encoder's trainable parameters are substantially fewer than those of the LoRA adapter in the LLM backbone during instruction tuning, and its parameter share of the overall framework is minimal for inference, underscoring its efficiency.

**Ablation Study of ECG Encoder** we conducted additional experiments comparing our default 1-D Temporal Convolution ECG encoder with alternative architectures, including: 1. S4-based Model: Vim-B (Vision Mamba, 98M parameters) (Zhu et al., 2024). 2. Transformer-based Model: ViT-B/16 (Vision Transformer, 86M parameters) (Dosovitskiy et al., 2020), adapted for 1-D token patching to align with the temporal nature of ECG signals. 3. SSL-Transformer Model: ViT-B/75 initialized with self-supervised learning (SSL) weights specific to ECG signals (Na et al., 2024). We evaluated these

---

[6]https://pytorch.org/docs/stable/generated/torch.nn.BatchNorm1d.html

Table 9: Comparisons of results with and without supervised manner. We take LLaMA-2-Instruct as the LLM backbone here. heavy teal color indicates the highest results.

| METHODS | SIZE | MIMC-IV-ECG | | | | PTB-XL | | | |
|---|---|---|---|---|---|---|---|---|---|
| | | BLEU-4 | METEOR | ROUGE-L | CIDEr-D | MTA | MTA | LC | DA |
| Vision Mamba | 86M | 0.548 | 0.737 | 0.715 | 5.58 | 3.78 | 3.88 | 3.61 | 3.50 |
| Vision Transformer | 98M | 0.592 | 0.815 | 0.772 | 5.67 | 4.33 | 4.15 | 4.12 | 3.78 |
| Vision Transformer (SSL) | 98M | 0.581 | 0.822 | 0.766 | 5.75 | 4.42 | 4.28 | 3.85 | 3.85 |
| 1-D Temporal Conv (Ours) | 20.4M | 0.610 | 0.799 | 0.773 | 5.78 | 4.52 | 4.38 | 4.01 | 3.98 |

models on two tasks: Quality of Generated Reports using the MIMC-IV-ECG dataset, and Evaluation of Alignment with Human Expert Annotations using the PTB-XL dataset. For fair comparison, we used Meta-Llama-3-8B-Instruct as the LLM backbone due to its consistent strong performance.

The results, summarized in the table below, show that our 1-D Temporal Convolution ECG encoder, despite having significantly fewer parameters, performs comparably or better across most metrics compared to ViT and ViT-SSL, and comprehensively outperforms the S4-based Vim. Notably, the ViT-SSL encoder demonstrates the benefit of self-supervised pretraining for initial ECG representation learning. However, our default ECG encoder effectively captures the 12-channel ECG temporal patterns while remaining lightweight, making it well-suited for our efficient instruction tuning framework. These findings validate the effectiveness of our 1-D Temporal Conv encoder and also provide valuable insights for future work, including designing more complex ViT-based architectures optimized for ECG time-series data.

Table 10: Parameter Comparison of ECG encoder and LLM backbone. We use LLaMA-1 7B as an example.

| MODULE | Trainable Params | Inference Params |
|---|---|---|
| LLM backbone | 159M | 6.90B |
| ECG encoder | 20.4M | 20.4M |

## A.3 FURTHER ANALYSIS OF GENERATED PROMPTS

**Prompts Number Analysis** In the ECG instruction data curation, we manually created 32 prompt examples, as illustrated in Section 3.3. To increase the diversity of our samples, we employed GPT-4 to rephrase these manually designed prompts, generating a larger pool of prompt examples. These generated examples were randomly sampled and paired with ECG-text pairs to compile the ECG instruction dataset. In this section, We compare the experiment's effects using 128, 256, and 512-generated samples, respectively. Table 11 shows the corresponding results with different dimensions. When the number is 256, it can achieve better results in most experimental settings. Hence, we take 256 generated samples as our default setting during the instruction tuning and inference.

Table 11: Performance comparison of different numbers of generated prompt samples. We evaluate them on the MIMIC-IV-ECG dataset, using BLEU-4, METEOR, ROUGE-L, and CIDEr-D metrics. We take LLaMA-1 7B as the LLM backbone here. heavy teal color indicates the highest results.

| PROMPT NUMS | BLEU-4 | METEOR | ROUGE-L | CIDEr-D |
|---|---|---|---|---|
| 128 | 0.541 | 0.756 | 0.718 | 5.15 |
| 256 | 0.543 | 0.761 | 0.724 | 5.26 |
| 512 | 0.538 | 0.754 | 0.732 | 5.03 |

**Ablation Study on GPT-4 Prompt Rephrasing** We also conducted an ablation study to compare the performance with and without GPT-4 rephrasing prompts, using a fixed prompt for the latter. The results in the following Table 12 indicate that using diverse prompts rephrased by GPT-4 leads to better performance, highlighting the superiority of instruction tuning in enhancing LLMs' generalization to new tasks and data over direct fine-tuning.

Table 12: Performance comparison of with and without GPT-4 prompt rephrasing. We take Mistral-Instruct as the LLM backbone here. heavy teal color indicates the highest results.

| PROMPT NUMS | BLEU-4 | METEOR | ROUGE-L | CIDEr-D |
|---|---|---|---|---|
| *w.o.* Rephrasing | 0.564 | 0.745 | 0.738 | 5.50 |
| *w.* Rephrasing (Ours) | 0.576 | 0.768 | 0.751 | 5.62 |

## A.4 COMPARISON WITH ENCODER-DECODER MODELS

In this section, we conducted additional comparative experiments using two open-source traditional encoder-decoder architectures: BART-Large (406M parameters) (Lewis, 2019) and T5-Large (780M parameters) (Raffel et al., 2020), as shown in Table 13. In adapting our framework for ECG instruction tuning, we employ the language encoder to process the input instruction, an ECG encoder to handle the input ECG signals, and the language decoder to generate the ECG report based on the output from both language end ECG encoder.

Our findings indicate that the performance of encoder-decoder models is comparable to the small pre-trained language models (GPT2-Medium and GPT-Large) presented in Table 1 and Table 2 of our paper. Moreover, LLM-based backbones (such as LLaMA1-2) consistently achieve a significant margin of improvement over the encoder-decoder architectures across all metrics.

Table 13: Comparison with encoder-decoder-based models on MIMIC-IV-ECG. For model size, 'M' denotes the million level, and 'B' denotes the billion level. The light teal color indicates the second highest results, and heavy teal color indicates the highest results.

| MODELS | SIZE | BLEU-1 | BLEU-2 | BLEU-3 | BLEU-4 | METEOR | ROUGE-L | ROUGE-1 | ROUGE-2 | CIDEr-D |
|---|---|---|---|---|---|---|---|---|---|---|
| BART-Large | 406M | 0.525 | 0.498 | 0.466 | 0.388 | 0.455 | 0.472 | 0.5124 | 0.451 | 3.15 |
| T5-Large | 780M | 0.595 | 0.542 | 0.465 | 0.422 | 0.498 | 0.456 | 0.522 | 0.438 | 4.08 |
| LLaMA-1 | 7B | 0.685 | 0.648 | 0.615 | 0.543 | 0.761 | 0.724 | 0.742 | 0.642 | 5.26 |
| LLaMA-2-Instruct | 7B | 0.706 | 0.662 | 0.622 | 0.581 | 0.775 | 0.745 | 0.768 | 0.664 | 5.55 |

## A.5 ANALYSIS OF COMBINING MEIT WITH A SUPERVISED MANNER

In this section, we conduct a new experiment where we trained a CNN (ECG encoder) in a supervised manner on the PTB-XL training set, utilizing all available annotations (approximately 70 patterns), as shown in Table 14. We then transferred the CNN for ECG instruction fine-tuning on both the MIMIC-IV-ECG and PTB-XL datasets. Our findings indicate that performance increased on the PTB-XL dataset in most metrics, likely due to the model's prior learning of specific annotated patterns. However, performance fluctuated on the MIMIC-IV-ECG dataset, which contains more data and exhibits greater diversity. This suggests that the supervised approach may enhance performance on in-domain data, but it limits generalizability to data from unseen domains.

Table 14: Comparisons of results with and without supervised manner. We take LLaMA-2-Instruct as the LLM backbone here. heavy teal color indicates the highest results.

| METHODS | PTB-XL | | | |
|---|---|---|---|---|
| | BLEU-4 | METEOR | ROUGE-L | CIDEr-D |
| MEIT | 0.439 | 0.675 | 0.594 | 4.05 |
| MEIT + Supervised manner | 0.445 | 0.664 | 0.612 | 4.12 |
| | MIMIC-IV-ECG | | | |
| | BLEU-4 | METEOR | ROUGE-L | CIDEr-D |
| MEIT | 0.581 | 0.775 | 0.745 | 5.55 |
| MEIT + Supervised manner | 0.578 | 0.778 | 0.739 | 5.47 |

## A.6 COMPUTATIONAL COST ANALYSIS OF MEIT

The time cost experiment, detailed in the Table 15, was conducted on the MIMIC-IV-ECG dataset. We found that larger models have longer training and inference times. Thus, we are considering techniques like quantization and other compression methods to improve model efficiency in future work.

Table 15: Computational time Analysis of MEIT with various parameters and backbones.

| MODEL | SIZE | Training time | Testing time |
|---|---|---|---|
| | | 4 A100 and 3 Epochs | 1 A100 and 128 Generated Samples |
| GPT-2 Large | 774M | 3.25h | 3.125min |
| LLaMA-2-Instruct | 7B | 13.5h | 9 min |
| LLaMA-2-Instruct (+) | 13B | 27h | 14.125 min |

### A.7 VISUALIZATION OF GENERATED ECG REPORT SAMPLES

As illustrated in Figures 8, 9, and 10 we have visualized the report samples generated by LLaMA-1, LLaMA-2, and Mistral-Instruct. The samples are presented in blue font to highlight the key information that aligns with the ground truth. The visualization demonstrates that all three selected models can capture the essential patterns of ECG signals and generate accurate reports. This underscores the efficacy of our proposed **MEIT** framework, which is adaptable to most open-source LLMs. It effectively learns the correct clinical semantics of ECG signals, thereby enabling the generation of corresponding reports.

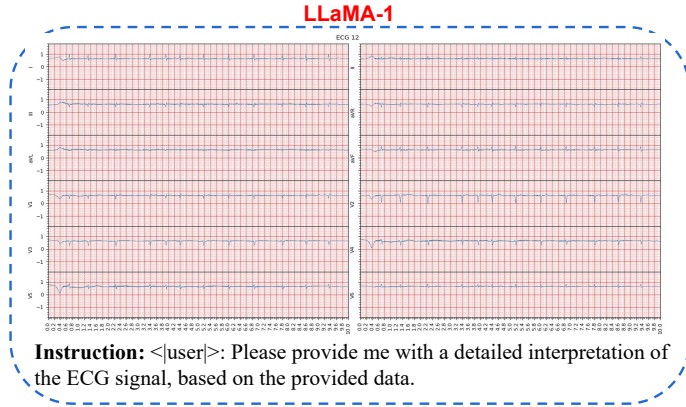

**Instruction:** <|user|>: Please provide me with a detailed interpretation of the ECG signal, based on the provided data.

**Generated Report**: <|assistant|>: atrial fibrillation. extensive st-t changes may be due to myocardial ischemia. abnormal ecg.

**Ground Truth**: <|assistant|>: atrial fibrillation. possible inferior infarct - age undetermined. possible anteroseptal infarct - age undetermined. lateral st-t changes may be due to myocardial ischemia. abnormal ecg.

Figure 8: Reports generated by LLaMA-1 following ECG instruction Tuning.

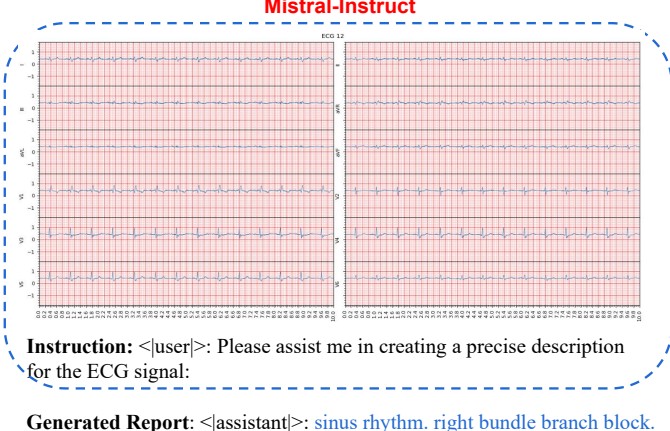

**Instruction:** <|user|>: Given the ECG signal embeddings, please help me generate an accurate description for this ECG signal embeddings:

**Generated Report**: <|assistant|>: sinus rhythm with 1st degree a-v block. left axis deviation. left ventricular hypertrophy. inferior/lateral st-t changes may be due to hypertrophy and/or ischemia. abnormal ecg. 

**Ground Truth**: <|assistant|>: sinus rhythm with 1st degree a-v block. left axis deviation. probable normal variant. inferior/lateral st-t changes may be due to myocardial ischemia. abnormal ecg. 

Figure 9: Reports generated by LLaMA-2 following ECG instruction Tuning.

**Instruction:** <|user|>: Please assist me in creating a precise description for the ECG signal:

**Generated Report**: <|assistant|>: sinus rhythm. right bundle branch block. abnormal ecg.

**Ground Truth**: <|assistant|>: sinus rhythm. indeterminate axis. right bundle branch block. abnormal ecg. 

Figure 10: Reports generated by Mistral-Instruct following ECG instruction Tuning.