# OpenReview forum: "ECG Instruction Tuning on Multimodal LLMs for Report Generation: Benchmark and Evaluation"
_ICLR.cc/2025/Conference — Submitted to ICLR 2025_

### Official Review · Reviewer_GHJv · 2024-11-02

**Soundness:** 4
**Presentation:** 3
**Contribution:** 3
**Rating:** 8
**Confidence:** 3

**Summary:**

This paper proposes a framework to automatically analyze ECG recordings and provide medical reporting using an LLM multimodal approach. Using two different sources of ECG signals and reports (MIMIC-IV-ECG and PTX-XL), the authors benchmark multiple open-source LLMs and compare them with several metrics, evaluating the accuracy, quality and fluency, as well as testing the models ability to perform zero-shot transferability, robustness to signal noise, and comparing the generated reports with the ones from experts.

**Strengths:**

1. The paper is original since it focuses on medical report generation of ECG signals, for which methods are not as developed as for images
2. The experiments were carefully designed, covering multiple LLM models and reporting insightful metrics for benchmarking and quality assessment. The results are thoroughly explained.
3. The paper is well-structured and explained, with detailed explanations and figures
4. This work can prove to be relevant in the medical field domain, and help guide future research on the topic

**Weaknesses:**

Some directions for improvement:
1. Although the results are detailed and commented on, the paper lacks some analysis of its limitations. It would be relevant to know for which cases the models cannot provide an accurate report so that improvements could be considered. Are there medical conditions that are underrepresented in the datasets? Or particular ECG biomarkers that the model has trouble with?
2. Although the model uses Gaussian noise to evaluate the robustness to signal perturbations, real-world noise present in ECG acquisitions is often more complex (e.g., baseline wander, motion artifacts, muscle artifacts...). Testing with more realistic signal perturbations could help understand the applicability to clinical settings.
3. The paper doesn't compare the performance of the model with other AI approaches that analyze and interpret ECG signals. Since the reports often consist of simple statements related to the rhythm and ECG waveforms, which are extensively covered in the literature by classification models, the real impact and advantage of the proposed framework is not fully assessed. How could the generated reports help in cases where the diagnosis is not so clear or that require further investigation?
4. Future directions are scarce and could be more specific. By recognizing the limitations of the approach (which are not clearly stated), there could be some pointers to study the feasibility and application in clinical environment, interpretability, improvement of its generalization ability, and integration of other types of data.

**Questions:**

1. Have you noticed any patterns in the test data that could provide insights into the strengths and weaknesses regarding the accuracy of the generated reports?
2. How could the generated reports help in cases where the diagnosis is not so clear or that require further investigation? Or what is the main advantage of generating ECG reports when there exist classification models for several heart conditions?
3. Although it would be interesting to test this framework for other signals such as EEG, what further work should be carried out to improve the current results? And does the performance of this approach match the state of the art methods for image data reporting?

---

### Official Review · Reviewer_NZRb · 2024-11-03

**Soundness:** 3
**Presentation:** 3
**Contribution:** 3
**Rating:** 6
**Confidence:** 4

**Summary:**

The authors propose a novel approach leveraging large language models (LLM) to automate the generation of diagnostic reports from ECG signals. Unlike conventional methods that focus solely on classifying signal anomalies, the proposed MEIT method aligns ECG signals with text generation through instruction tuning. The performance of the model is validated across multiple aspects, including report generation quality, zero-shot capability, noise robustness, and expert comparison, using two extensive ECG databases.

**Strengths:**

The MEIT framework innovatively aligns the ECG signal processing and report generation process through an instruction tuning approach, enabling LLMs to generate diagnostic reports. This represents a rare and significant advancement in the domain of automated ECG diagnosis.

The authors conducted comprehensive validation using well-established metrics and provided an extensive analysis of the model's stability and generalization performance, clearly demonstrating the feasibility of the proposed method.

**Weaknesses:**

The authors focus heavily on comparisons involving instruction tuning of the language model, raising concerns about whether the chosen ECG encoder is an effective architecture. It would be beneficial to evaluate the use of pre-trained state-of-the-art models as the ECG encoder to ensure robustness.

In Section 5.2.4, the authors mention a comparison with 500 ground-truth reports but should clarify the source of these reports. Are they confirmed to be manually annotated by physicians? Additionally, how many types of abnormal ECG events are represented within these 500 reports? For a dataset like PTBXL, which includes 71 abnormal categories, having only 500 reports would mean at most seven data points per category, which may not be sufficient for thorough validation.

**Questions:**

In the design of zero-shot experiments, the authors assume that training on MIMIC-IV and testing on PTBXL accounts for differences in population data and collection devices between databases. However, ECGs follow fixed patterns, and diagnostic criteria are globally standardized if these pattern changes are accurately captured. Moreover, there is no assurance that the training set from MIMIC-IV does not include abnormalities present in PTBXL. Given that PTBXL covers 71 types of abnormalities, many are likely represented in the MIMIC-IV training data. This questions the validity of the approach as true zero-shot learning.

---

### Official Review · Reviewer_E4Mu · 2024-11-03

**Soundness:** 3
**Presentation:** 3
**Contribution:** 2
**Rating:** 5
**Confidence:** 4

**Summary:**

The paper presents the Multimodal ECG Instruction Tuning (MEIT) framework, designed to automate ECG report generation using large language models (LLMs) and multimodal instruction tuning. The main contribution is alignment of ECG signals with corresponding text descriptions to streamline report generation, across two popular ECG datasets (MIMIC-IV-ECG and PTB-XL). The authors present results using nine open-source LLMs. The results demonstrate the effectiveness of the proposed approach in generating quality report generation, zero-shot capabilities, resilience to signal perturbation, and alignment with human expert evaluation. The evaluation includes metrics like BLEU and ROUGE and aligns with human expert assessments making it a comprehensive evaluation.

**Strengths:**

The paper is well-written and easy to understand. The authors maintain coherence and fluency throughout the paper with minimal language errors. The contribution of this work is of significant interest to the community but the novelty is incremental. The authors do a great job in my opinion in formulating the problem and the instruction for text data and a Comprehensive evaluation through showcasing their proficiency in quality report generation, zero-shot capabilities, resilience to signal perturbation, and alignment with human expert evaluation.
Additionally, the proposed alignment approach aids in addressing the catastrophic forgetting of general knowledge.

**Weaknesses:**

The contribution is marginal at best with primarily being conversion of ECG-text pairs into a chat-bot style instruction format for facilitation self-attention based learning between the ECG and text embeddings. Literature review can be better and more comprehensive. There is decent body of work on ECG diagnosis and report generation with LLM that has not been cited. Ex:
1. Yu, Han, Peikun Guo, and Akane Sano. "Zero-shot ECG diagnosis with large language models and retrieval-augmented generation." Machine Learning for Health (ML4H). PMLR, 2023.
2. Yu, Han, Peikun Guo, and Akane Sano. "ECG Semantic Integrator (ESI): A Foundation ECG Model Pretrained with LLM-Enhanced Cardiological Text." arXiv preprint arXiv:2405.19366 (2024).
Line 194 is grammatically incorrect.

**Questions:**

Why is it important to handle catastrophic forgetting of general knowledge and not focus solely on the task of ECG report generation if we are training the model? Models can be specific to tasks and that should not be a problem that needs to be addressed with additional overhead.

ECG text data is vague, do you mean SCP statements in particular? SCP statements are the textual data that provide the information and technical details of ECG signals, however some of the features noted in SCP statements are not merely language details and require reasoning and interpretation of the signal Ex: computing the axis deviation for a given signal and reporting it. Trusting an LLM on these values of SCP statements would be questionable without explaining and evidence into how these are generated.

---

### Official Review · Reviewer_8Eqp · 2024-11-04

**Soundness:** 2
**Presentation:** 3
**Contribution:** 2
**Rating:** 3
**Confidence:** 5

**Summary:**

This work presents a multi-modal framework to generate ECG reports from ECG signals by instruction tuning LLMs on paired ECG ECG and reports datasets. To add a signal modality to the LLMs, the authors utilize an ECG encoder composed of 1D convolutional layers, and manipulate the LLMs to fuse the ECG embeddings in a self-attention stage. The authors evaluate 12 different LLMs with 2 different datasets (MIMIC-IV-ECG and PTB-XL) on the report generation tasks using some conventional metrics (BLEU, ROUGE, etc) to compare the generated reports with the ground truth reports.

**Strengths:**

* The paper is well written.
* The authors have conducted an extensive set of experiments with various LLMs combined with ECG signals, which is quite impactful since combining LLMs with physiological signals is not yet explored enough in this field.

**Weaknesses:**

* The motivation of this work is not convincing enough. Specifically, the authors do not explain why the report generation task in the ECG domain is needed and how the proposed task can be applied to medical practice, which is very important in medical field. The paired reports in MIMIC-IV-ECG or PTB-XL used in this work are mostly composed of keyword-based statements (e.g., "Normal ECG", "Sinus Rhythm", etc.), which is automatically generated from built-in algorithms from ECG machines. Therefore, if models are trained in a supervised manner using these reports as ground truths, they just do an approximation of the algorithms, which seems not meaningful at all to my understanding.
* The authors repeatedly state their framework enables the LLMs to generate professional-grade reports, which seems an overclaim. Although the authors show the alignment scores between LLM-generated reports and human-written reports on the 500 samples from the PTB-XL dataset, they still should show alignment scores between the ground truth reports (which has been used to train the LLMs) and human-written reports as a baseline as well.

**Questions:**

* Can you present the alignment scores between ground truth reports and human-written reports, and analyze the results? If the ground truth reports show high alignment with human-written reports, why do we train the model instead of directly using these reports given that the reports already can be acquired from the automatic generation algorithms from ECG machines? If not, how can we trust the model trained in a supervised manner with these "unaligned" reports?

---

### Official Review · Reviewer_zL6Y · 2024-11-04

**Soundness:** 2
**Presentation:** 2
**Contribution:** 3
**Rating:** 5
**Confidence:** 4

**Summary:**

This paper proposes the MEIT framework, which is the first framework to enable ECG Instruction Tuning with LLM for the downstream task of ECG report generation. Within the framework, the authors also introduce the lightweight concatenated-fusion strategy to align the ECG and text modalities together. Finally, the authors also propose a benchmark that aims to assess the generated reports from the MEIT framework with various evaluation methods. The evaluation methods range from assessing the quality of the generated reports with conventional NLG metrics such as BLEU and METEOR to zero-shot generalizability, signal robustness, and comparison to human expert annotations using GPT-4o.

**Strengths:**

1. The challenge of ECG Instruction Tuning compared to images for multimodal LLMs in the introduction and medical report generation, instruction tuning, and LLMs for ECG in the related works are well-written and organized.
2. Many LLMs (~12 LLMs) ranging from GPT-2 models to more recent LLaMA-3-Instruct models are compared in the experiments section to assess the quality of report generation.
3. The authors tackle an important area of research that has not been explored yet like the image-text multimodal domain.

**Weaknesses:**

1. Of the three parts in the MEIT framework which are the ECG encoder, modality alignment, and LLM backbone, the ECG encoder only uses several 1-D CNN layers and average pooling. There are many ECG-specific architectures using SSL or other transformer-based architectures that can significantly improve the current ECG encoder. Some of these architectures should be compared and explored similar to how different methods were explored for the other two parts of the framework.
2. For the Signal Perturbation Robustness task in the established benchmark, adding noise to ECG signals can potentially change the correct ground-truth report because ECG is very sensitive to noise, and there are many noise categories such as baseline wander and drift for ECG. Therefore, I am not sure if the robustness analysis is appropriate in this multimodal LLM ECG report generation setting.
3. There are no specific reasons stated for conducting ablation experiments on a subset of the total number of LLMs used in report generation quality comparison. For example, BLOOM, OPT, LLaMA-1, Mistral were used for robustness analysis, LLaMA-2-Instruct and LLaMA-3-Instruct were utilized for evaluation of alignment with human expert annotations. However, there were no intuitive insights mentioned in the paper for these selections.
4. There are many minor grammatical errors and typos in the paper.
In line 63 MELT should be MEIT
Line 160 should be 800k
Line 191 that require should be requires, and via directly injects should be revised in Line 194

**Questions:**

1. Why was the simple 1-D CNN layer architecture with average pooling selected for the ECG encoder?
2. As mentioned in the weaknesses, what is the intuition behind the selected models for the robustness analysis and the evaluation of alignment with human expert annotations? Also, who specifically are "human medical experts" mentioned in section 5.2.4?
3. What is the reason or intuition for lightweight attention-based concatenated-fusion outperforming LLaVa and Flamingo alignment methods?

---

### Official Review · Reviewer_DCvH · 2024-11-06

**Soundness:** 3
**Presentation:** 3
**Contribution:** 3
**Rating:** 6
**Confidence:** 3

**Summary:**

The authors introduce a new approach for generating ECG reports from ECG signals. They present MEIT, a technique that fine-tunes the best-performing large language models (LLMs) through instruction tuning. This process uses representations from an ECG encoder, which is trained simultaneously with the fine-tuning of the LLM. Furthermore, due to the lack of relevant existing literature, the authors propose a benchmark using existing databases such as MIMIC or PTB-XL.

**Strengths:**

-  As discussed in the manuscript, the automatic generation of medical reports is a useful tool with severe real word applications.
- The authors have conducted an extensive evaluation involving multiple LLM models.
- The results show that MEIT is able to perform the task for which it has been designed with remarkable performance.

**Weaknesses:**

- In my opinion, there is a lack of context as to how the proposed method differs from the different Instruction tuning methods for computer vision, mentioned in line 82. Although I understand that the objective is different (between explaining an image and generating a medical report) I consider that the optimization method is similar in both cases and I would like to know what makes MEIT a special optimization method compared to MiniGPT-4 or the others mentioned.

- Linked to the above, the authors point out that, due to the particularities of ECG signals, these Computer Vision methods are not applicable. However, the authors propose an ECG encoder that is based on Convolutional Layers, which is a commonly used architecture in Computer Vision, so it is not very clear to me what makes these methods inapplicable in this particular context. IMO, if some of these computer vision methods can be applied to the proposed framework, they should serve as baselines for MEIT evaluation

- Although the authors comprehensively evaluate different LLM models, they only use one ECG encoder. I believe that using other architectures such as Transformers or S4 models, which are also used to process cardiac signals, could improve the results.

**Questions:**

- My only question is about creating medical reports from the PTB-XL database. To my knowledge, this database does not include ECG reports, just tabular data. If these reports have been created using these labels, could the authors detail how the proposed method would differ from using a classification model that infers these labels and creating a medical report based on them in the same way that the GT report has been created for comparison?

---

### Meta-Review · Area_Chair_dGCi · 2024-12-21

**Metareview:**

The paper introduces MEIT, a multimodal framework using instruction-tuned LLMs for ECG report generation, evaluated on two datasets, showcasing good performance, clinical relevance, and potential real-world application.

Strengths
- MEIT introduces a novel approach for automating ECG report generation with LLMs, demonstrating strong zero-shot capabilities and robustness to signal perturbation
- The paper conducts extensive experiments with diverse LLMs and multiple benchmarks

Weaknesses
- The paper fails to justify the need for generating ECG reports when machine-generated reports already exist and lacks evidence of its practicality in ICU and home care settings with reduced-lead ECGs.
- The paper's main contributions are incremental, focusing on adapting existing LLMs for ECG data without significant methodological innovation, such as relying on standard 1D CNNs for the ECG encoder.
- The robustness analysis is incomplete, focusing solely on Gaussian noise while omitting critical real-world noise types like baseline wander and drift

**Additional Comments On Reviewer Discussion:**

During the rebuttal, reviewers raised concerns regarding the lack of care put into the ECG encoder design, which the authors failed to address adequately. Most importantly, the authors failed to justify the contribution of this work; 12-lead ECG machines can already automatically generate reports, therefore this work is redundant. Reduced-lead ECG machines (typically used for home care) might not be equipped with such functionality, but the proposed work has never demonstrated its effectiveness in reduced-lead scenarios anyway.

---

### Decision · Program_Chairs · 2025-01-22

Reject